# Immunophenotyping of Peripheral Blood Cells in Patients with Chronic Lymphocytic Leukemia Treated with Ibrutinib

**DOI:** 10.3390/cells13171458

**Published:** 2024-08-30

**Authors:** Pierre Stéphan, Khaled Bouherrou, Yann Guillermin, Anne-Sophie Michallet, Yenkel Grinberg-Bleyer

**Affiliations:** 1Cancer Research Center of Lyon, UMR INSERM 1052, CNRS 5286, Université Claude Bernard Lyon 1, Labex DEV2CAN, Centre Léon Bérard, 69008 Lyon, France; pierre.stephan@aphp.fr (P.S.); khaled.bouherrou@lyon.unicancer.fr (K.B.); 2Hematology Department-Centre Léon Bérard, 69008 Lyon, France; yann.guillermin@lyon.unicancer.fr (Y.G.); anne-sophie.michallet@lyon.unicancer.fr (A.-S.M.)

**Keywords:** chronic lymphocytic leukemia, onco-immunology, immunotherapy, spectral flow cytometry

## Abstract

Chronic lymphocytic leukemia (CLL) is a B-cell-derived hematologic malignancy whose progression depends on active B-cell receptor (BCR) signaling. Despite the spectacular efficacy of Ibrutinib, an irreversible inhibitor of Bruton tyrosine kinase (BTK), resistance can develop in CLL patients, and alternative therapeutic strategies are therefore required. Cancer immunotherapy has revolutionized cancer care and may be an attractive approach in this context. We speculated that characterizing the immune responses of CLL patients may highlight putative immunotherapeutic targets. Here, we used high-dimensional spectral flow cytometry to compare the distribution and phenotype of non-B-cell immune populations in the circulating blood of CLL patients treated with Ibrutinib displaying a complete response or secondary progression. Although no drastic changes were observed in the composition of their immune subsets, the Ibrutinib-resistant group showed increased cycling of CD8+ T cells, leading to their overabundance at the expense of dendritic cells. In addition, the expression of 11 different surface checkpoints was similar regardless of response status. Together, this suggests that CLL relapse upon Ibrutinib treatment may not lead to major alterations in the peripheral immune response.

## 1. Introduction

Numerous studies have demonstrated the close link between the immune response to cancer and its outcome. Our understanding of the mechanisms underlying tumor escape from the immune system has culminated with the development of immunotherapies targeting the inhibitory receptors expressed by T lymphocytes, named “checkpoint-blockade therapies” [1]. Antibodies directed at CTLA-4, PD (L)-1 and LAG-3, which notably reinvigorate “exhausted” CD8^+^ T cells, are highly effective in some solid tumors and hematologic malignancies, although the majority of patients display primary or secondary resistance. The identification of immune alterations (e.g., the aberrant expression of immune checkpoints) in the tumor microenvironment has offered new therapeutic perspectives for treating cancer.

Chronic lymphocytic leukemia (CLL), the most common hematologic malignancy in Western countries, is characterized by the accumulation of mature B cells in lymphoid tissues and in the blood [2]. A hallmark of CLL is a perturbed immune profile, affecting the T-cell compartment in particular [3].

CD4^+^ T cells are highly activated and accumulate in the blood of CLL patients, where they are believed to promote disease progression, for instance, through the expression of IL-4 and IL-21 [4]. Though several studies have shown increased CD4^+^Foxp3^+^ regulatory T cells (Treg) in CLL patients, their prognostic value is under debate [5,6]. CD8^+^ T cells express high levels of PD-1 and other inhibitory receptors, suggesting a state of exhaustion, confirmed by their impaired cytotoxic function in vitro [7,8,9]. Similarly, natural killer (NK) cells seem to be less functional in CLL samples [10,11], albeit they are more abundant in the bloodstream [12]. Overall, this is suggestive of impaired anti-tumor immunity that could be harnessed by immunotherapy. 

CLL progression largely relies on intense B-cell receptor (BCR) signaling, and small molecule inhibitors have successfully been used to target the BCR pathway, notably Ibrutinib, which inhibits Bruton tyrosine kinase (BTK) [13]. Though this inhibition leads to high response rates in patients, secondary resistance (relapse) is frequent [14]. In addition to its impact on malignant and normal B cells, Ibrutinib has been shown to perturb other immune subsets. Indeed, through its partial inhibition of interleukin-2-inducible kinase (ITK), the drug reduces the absolute counts of both CD4^+^ and CD8^+^ T cells [15,16] but interestingly may improve the ex vivo cytotoxic function of the latter against CLL cells [17]. In this context, characterizing the phenotype of the immune cells in patients responding or not responding to Ibrutinib may not only provide a mechanistic understanding of CLL’s progression but also highlight potentially novel immunotherapeutic strategies. In this study, we performed deep phenotyping of the immune cells in the circulating blood of CLL patients using spectral flow cytometry.

## 2. Materials and Methods

### 2.1. Samples for Spectral Flow Cytometry Analyses

Blood samples from CLL patients with a long-lasting response to Ibrutinib (n = 10) or undergoing relapse (n = 8) were obtained under informed consent from the Hematology Department of the Centre Léon Bérard, Lyon, France. Their characteristics are presented in Table 1 and Table 2. The procedure was approved by the local clinical trial review committee according to the national ethical recommendations and conducted in compliance with the Declaration of Helsinki. Patient response or relapse was established based on their blood counts and clinical examination according to the iwCLL guidelines [18]. 

### 2.2. Flow Cytometry

Peripheral blood mononuclear cells (PBMCs) were isolated by Ficoll density gradient centrifugation, and red blood cells were lysed with Ammonium Chloride–Potassium lysis buffer. The cell suspensions were frozen in 90% FBS–10% DMSO buffer. After thawing in a water bath at 37 °C, the samples were washed in RPMI 1640 W/HEPES W/GLUTAMAX-I (supplemented with 10% FBS; Penicillin/Streptomycin; Non-Essential Amino Acids; Sodium Pyruvate; and β-Mercaptoethanol, all from Thermo Fisher Scientific, Waltham, MA, USA). B cells were negatively selected by labeling the cell suspensions with a biotinylated anti-CD19 antibody (Biolegend, San Diego, CA, USA), followed by incubation with anti-biotin microbeads (Miltenyi Biotec, Tokyo, Japan). The cells were placed on a MojoSort™ Magnet (Biolegend, San Diego, CA, USA), and the unlabeled suspension was collected. A total of 1 million cells per sample were stained with a viability dye for 15 min at room temperature (RT) and then incubated with human Fc block for 10 min at RT. The cells were then incubated with surface marker antibodies mixed in FACS buffer and 10% brilliant stain buffer (BD) for 30 min at RT in the dark. The cells were then fixed and permeabilized using the eBioscience Foxp3/Transcription Factor Staining Buffer Set (Thermo Fisher Scientific, Waltham, MA, USA) according to the manufacturer’s instructions. Finally, the cells were incubated with a mix of intracellular antibodies for 20 min at 4 °C. The complete list of antibodies and their final concentrations are based on previous work [19] and are presented in Appendix A. Acquisition was performed on an Aurora spectral flow cytometer (Cytek, Fremont, CA, USA). Analyses were carried out using the OMIQ software (Dotmatics, Bishop’s Stortford, UK) (www.omiq.ai, accessed on 20 June 2023). 

### 2.3. Statistical Analyses

Statistics were calculated using GraphPad Prism Software v9 and R version 4.2.1. For the FACS data, Mann–Whitney tests were used, unless mentioned otherwise. Only statistically significant values are shown. Statistical analyses for comparison between clusters, established by the FlowSOM algorithm, were carried out using the edgeR package.

## 3. Results

### 3.1. Increased CD8^+^ T-Cell Frequencies upon Ibrutinib Relapse

To investigate the impact of response or relapse to Ibrutinib on peripheral immune subsets, B-cell-depleted PBMCs from 18 patients (10 response, 8 relapse) were phenotyped using spectral flow cytometry. We analyzed the major immune cell populations and their expression of various markers of differentiation, activation and exhaustion. We initially concatenated all the samples and performed an integrative analysis of the general immune subsets, using CD3, CD4, CD8, TCRgd, CD56, CD11c and CD163 as the variables. Unsupervised analysis through 2D reduction using UMAP and clustering with FlowSOM revealed nine major populations, including dendritic cells (DCs), natural killer (NK) cells and T-cell subsets (Figure 1A,B). We detected an increase in the proportion of CD8^+^ T cells at the expense of DCs in the patients undergoing treatment relapse, while all the other lineages remained unchanged (Figure 1C). This observation was confirmed using supervised gating analyses (Figure 1D). In line with this, the proportion of cycling (Ki67^+^) CD8+ T cells was significantly higher in patients following relapse (Figure 1E), suggesting subtle alterations in the peripheral immune profiles of patients relapsing after Ibrutinib treatment. 

To analyze whether this expansion of CD8^+^ T cells was associated with further modifications in their activation status, we examined their profiles using 17 additional variables, including maturation, activation and inhibition. A total of 20 clusters were identified using FlowSOM, which comprised naïve, cycling and effector cells (Figure 2A,B). Surprisingly, the expression of activating and inhibitory checkpoints was very heterogeneous, and many of the cell clusters expressed one or two checkpoints at most. Only two clusters were differentially represented between the responding and relapsing patients, with an increase in cycling Ki67^+^ CD8^+^ T cells upon relapse, confirming our supervised analyses (Figure 1E and Figure 2C). A cluster of CD45RA^+^TCF1^+^CD7^+^ cells was enriched in the responders, but the total proportion of CD45RA^+^ and CD45RA^+^TCF1^+^ cells upon manual gating was unchanged between groups (Figure 2D). Similar conclusions were reached for EOMES^+^ and GZMB^+^ cells (Figure 2D). Of note, the total proportion of PD-1/TIM-3-co-expressing CD8^+^ T cells, reported to represent exhausted T cells [20], was low in both groups. Boolean analysis confirmed the small number of cells expressing more than one inhibitory receptor (Figure 2E). This suggested that T-cell exhaustion was not a general feature of the circulating CD8^+^ T cells in CLL, at least not through the analysis of traditional inhibitory checkpoints. To further explore the impact of Ibrutinib treatment, we examined the correlation between the duration of Ibrutinib treatment and the phenotype of the CD8^+^ T cells. Interestingly, we detected an inverse correlation between the expression of PD-1 and CD39 and the duration of treatment in responding patients (Figure 2F,G). Conversely, LAG-3 expression was, surprisingly, enhanced in a time-dependent manner (Figure 2H). Together, this suggested a time-dependent effect of Ibrutinib on T-cell phenotype. Importantly, these correlations were observed solely in the responders, indicating that they may be the result of a return to homeostasis consecutive to reduced or abolished tumor burden. Functional in vitro studies would be required to further explore the impact of therapy on T-cell function. 

### 3.2. Impact of the Response Status on CD4^+^ T-Cell Subsets

Next, we explored the phenotype of the CD4^+^ T cells, starting with Foxp3- conventional T cells (Tconv cells). We applied the same strategy as for the CD8^+^ T cells and identified 21 cell clusters, including naïve cells, T-Bet^+^ TH1-like cells, and TFH cells, as well as three GZMB^+^ cytotoxic-like clusters (Figure 3A,B). Slight differences between groups could be noted, particularly an increased representation of ICOS^+^PD-1^+^ TFH-like cells in the relapsing patients, which was confirmed through the supervised analyses (Figure 3C,D). Although a cluster of GZMB^+^ cells was more abundant and a cluster of CD45RA^+^ naïve cells was less abundant upon relapse, this did not impact the overall proportion of cytotoxic and naïve CD4^+^ Tconv cells (Figure 3D). Thus, patients developing resistance to Ibrutinib were characterized by an increased fraction of TFH cells, which may have influenced disease pathophysiology. 

Treg cells are largely associated with deleterious effects in cancer through their inhibition of anti-tumor immunity. In CLL patients, we observed similar proportions of Treg cells within the CD4^+^ T cells regardless of patient response to Ibrutinib (Figure 4A). Unsupervised clustering revealed only four clusters, including a cluster of CD45RA^+^TCF-1^+^ naïve-like cells that was enriched in the relapsing patients, leading to a trend towards fewer CD4RA^+^ cells (Figure 4B–E), whereas the other Treg cell populations were unchanged. Hence, secondary resistance to treatment did not significantly impact the Treg cell compartment. 

### 3.3. Study of NK Cells in CLL Patients

In addition to the T-cell subsets, we also evaluated the distribution of NK cell subsets based on their expression of CD56, CD57, GZMB and KI67. As expected, CD56^bright^ and CD56^dim^CD57^+^ clusters could be identified through the unsupervised analysis. Most of the cells expressed GZMB, and a minority of cells that were cycling were found in these two main subsets (Figure 5A,B). Nevertheless, no statistical difference between groups could be detected, and similar conclusions were reached through manual gating (Figure 5C,D). Thus, NK cells were not significantly altered in patients displaying resistance to Ibrutinib.

### 3.4. Evaluation of Checkpoint Receptors on T and NK Cells

Antibodies targeting PD-1, CTLA-4 and more recently LAG-3 have improved cancer outcomes in many settings. However, investigations on their use in CLL patients are scarce [21,22,23,24]. We reasoned that checkpoint inhibition could be an interesting therapeutic avenue for CLL patients undergoing relapse. We thus compared the expression of 10 immune checkpoints, as well as CD39, in T and NK cells through traditional manual gating (Figure 6). Among the inhibitory checkpoints, PD-1 was broadly expressed by all subsets except the NK cells, while TIM-3 and NKG2A were mainly expressed by the latter. TIGIT^+^ cells were detected among all populations at a >20% frequency, while LAG-3^+^ cells were quite rare. CTLA-4 and CD39 expression was restricted to the Treg cells. Regarding activation receptors, OX-40 and 4-1BB expression could be noted in about half of the patients. ICOS^+^ cells were found only within the Treg and Tconv cells. Surprisingly, TNFR2 was solely detected on a minority of NK cells. Interestingly, NKG2A^+^ NK cells were less abundant in the relapsing patients while their expression of OX-40 was higher, suggesting improved function. The other subsets and receptors were unchanged between groups. Altogether, resistance to Ibrutinib was not associated with major alterations in the expression of druggable checkpoints.

## 4. Discussion

Although the impact of Ibrutinib therapy on both malignant B cells and normal immune cells is well described, it is unclear whether acquired resistance to the treatment is associated with alteration of the TME [25]. Given our growing understanding of the close link between cancer and immunity and the revolution brought about by immunotherapies, it seemed essential to decipher the putative impact of resistance to therapy on the normal immune compartments. 

Our spectral flow cytometry analyses highlighted a series of alterations in the circulating immune subsets. In particular, the fraction of cycling CD8+ T cells was significantly higher in the relapsing patients. We observed a reduction in naïve cell clusters in all the T-cell subsets. This suggests a rebound in the activation of tumor-specific immune cells due to the awaking of tumor burden, which had thus far been kept in check by treatment. To fully confirm this hypothesis, evaluating the clonality of the different T-cell populations would be insightful. Importantly, despite this increased CD8+ T-cell activation upon relapse, we did not detect significant changes in the expression of inhibitory receptors, indicating that T-cell exhaustion, a feature of CLL rescued by Ibrutinib [15,26], was not (yet?) impacted by secondary progression. This was in accordance with a recent study using scRNA-seq on a single patient [27]. However, importantly, most of the samples obtained from the relapsing patients were obtained quickly after a relapse diagnosis (Table 2); it would therefore be relevant to investigate the immune phenotype at a later time point. In addition, functional assays measuring cell proliferation, cytokine expression and function are needed to draw any definitive conclusions. 

Interestingly, within the CD4^+^ T-cell compartment, we detected a significant increase in PD-1^+^ICOS^+^ TFH-like cells in the progressing patients. Through their secretion of IL-21, TFH cells promote malignant B-cell proliferation and survival [28], and their proportions among PBMCs are correlated with tumor load. Our data suggest that the re-emergence of CLL may promote TFH cell polarization or expansion, which is consistent with the published data [29]. Considering these results, immunotherapeutic targeting of TFH cells, for instance, through anti-ICOS mAbs, currently tested in other hematologic malignancies [30], is a promising perspective.

Though BCL-2 inhibitors such as Venetoclax show an interesting response rate in these patients [31], new treatments are required for CLL patients experiencing primary or secondary resistance to Ibrutinib. Beyond chimeric antigen receptor (CAR)T cells [32], immunotherapies targeting the surface receptors expressed by non-malignant cells emerge as interesting treatment strategies. Thus, one main objective of our study was to identify checkpoints whose expression was up- or downregulated in relapsing patients. Such analyses may not only reveal immune biomarkers of relapse and fundamental mechanisms of anti-tumor immunity but also important targets. Nevertheless, none of the 11 tested markers displayed changed expression on the T-cell subsets, with only a minor reduction in NKG2A^+^ and increase in OX-40^+^ NK cells. This obviously does not preclude the assessment of the therapeutic potential of checkpoint inhibitors or agonist antibodies. 

Several limitations can be detected in our study. First, this study was restricted to a small sample series, which may have hindered possible differences between response statuses. Indeed, we limited our analyses to the circulating blood, and it would also be helpful to analyze the bone marrow and lymph node microenvironments. In fact, the minor changes in the immune parameters in the blood following treatment relapse might be much more pronounced in lymphoid tissues. In addition, we compared different series of patients undergoing complete response to treatment or relapse. To overcome the inter-patient heterogeneity detected in many of the parameters across our study, it would be insightful to perform a longitudinal study. Finally, although spectral flow cytometry is a powerful tool for the simultaneous assessment of many immune parameters, functional studies, for instance, assessing T-cell cytotoxicity, cytokine production, or exhaustion, are warranted to draw conclusions on the impact of therapeutic failure on the immune response.

## Figures and Tables

**Figure 1 cells-13-01458-f001:**
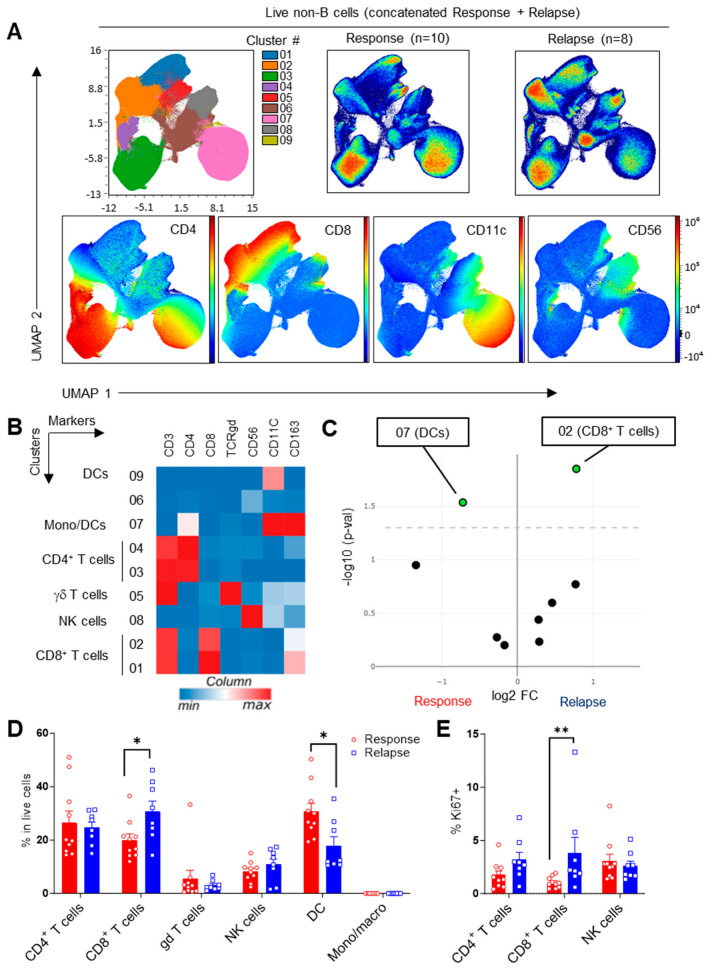
Immune subsets in CLL patients under Ibrutinib therapy. Samples were stained for FACS and analyzed using unsupervised clustering (**A**–**C**) and traditional supervised analyses (**D**,**E**) after manual gating on, B-cell-depleted live cells. (**A**) Uniform Manifold Approximation and Projection (UMAP) visualization, FlowSOM distribution of clusters and projection of selected markers in concatenated samples. (**B**) Heatmap showing hierarchical clustering and expression of indicated markers in FlowSOM clusters. (**C**) Volcano plot showing differential cluster enrichment. (**D**) Proportion of immune subsets using a supervised gating strategy. CD4^+^ T cells: CD3^+^TCRγδ^−^CD4^+^CD8^−^; CD8^+^ T cells: CD3^+^TCRγδ^−^CD4^−^CD8^+^; gd T cells: CD3^+^TCRγδ^+^; NK cells: CD3^−^CD56^+^CD7^+^; DCs: CD3^−^CD11c^+^CD7^−^; monocytes/macrophages: CD3^−^CD11c^−^CD163^+^. (**E**) Proportion of proliferating Ki67^+^ cells across cell populations. Means +/− SEM are shown; each dot represents a sample. Mann–Whitney tests were used. * *p* < 0.05, ** *p* < 0.005.

**Figure 2 cells-13-01458-f002:**
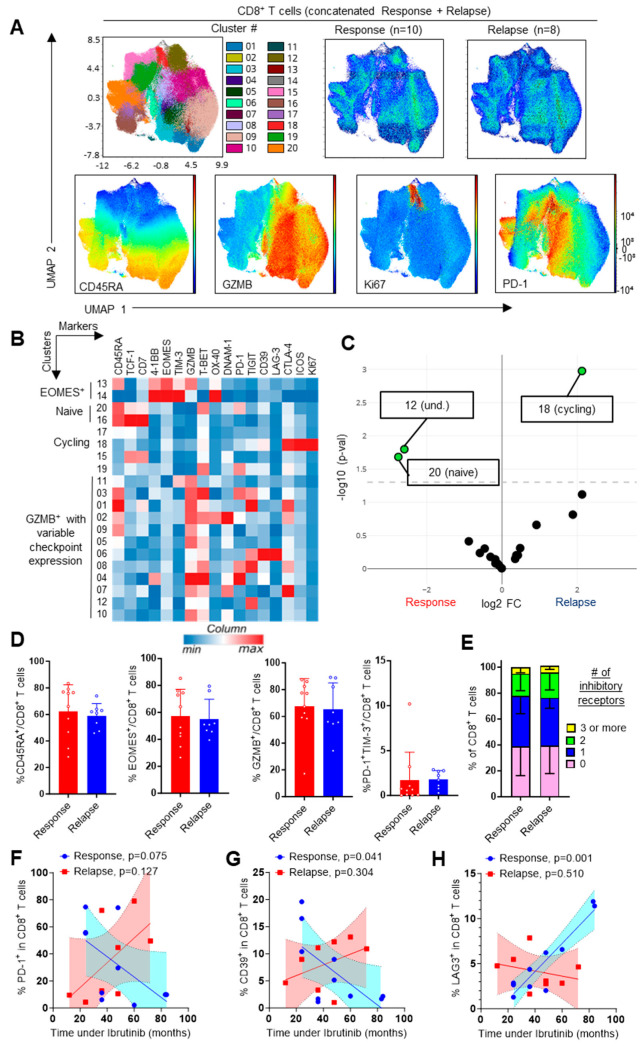
Phenotype of CD8^+^ T cells. (**A**) UMAP visualization, FlowSOM distribution of clusters and projection of selected markers in concatenated samples following manual gating on live CD8^+^ T cells. (**B**) Heatmap showing hierarchical clustering and expression of indicated markers in FlowSOM clusters. (**C**) Volcano plot showing differential cluster enrichment (und.: undefined). (**D**,**E**) Expression of the indicated markers (**D**) and Boolean analysis of inhibitory-checkpoint-expressing cells (**E**) upon manual gating. Means +/− SEM are shown; each dot represents a sample. Mann–Whitney tests were used. No statistical difference was detected. (**F**–**H**) Time-dependent impact of Ibrutinib therapy on the expression of PD-1, CD39 and LAG-3. Linear regression slopes, 95% confidence areas and *p*-values are shown.

**Figure 3 cells-13-01458-f003:**
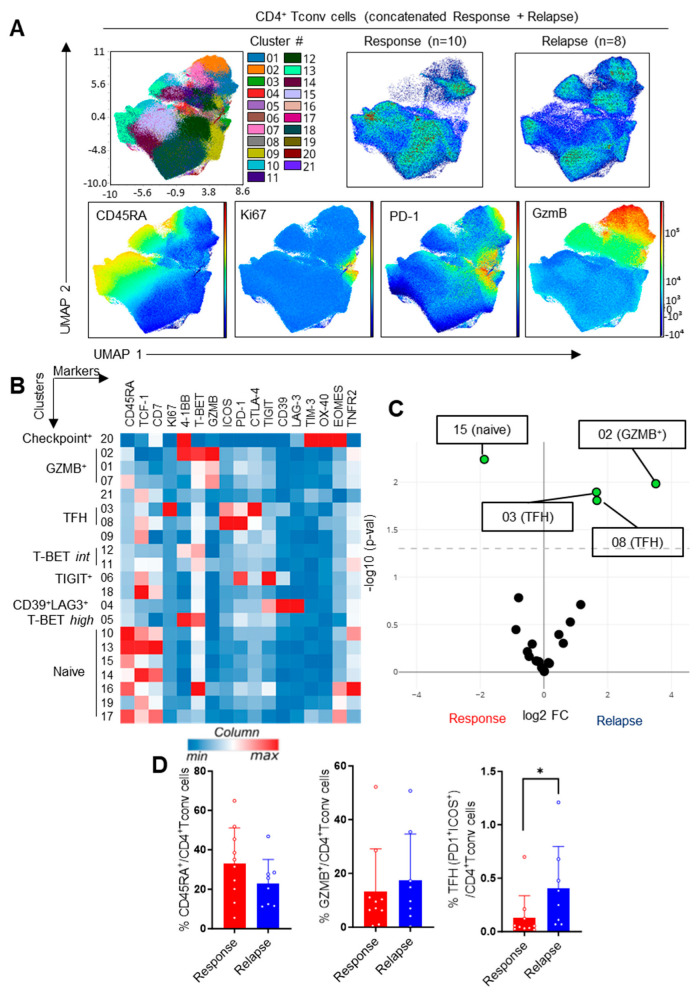
Analysis of CD4^+^ Tconv cells. (**A**) UMAP visualization, FlowSOM distribution of clusters and projection of selected markers in concatenated samples following manual gating on live CD4^+^Foxp3^−^ Tconv cells. (**B**) Heatmap showing hierarchical clustering and expression of indicated markers in FlowSOM clusters. (**C**) Volcano plot showing differential cluster enrichment. (**D**) Expression of the indicated markers upon manual gating. Means +/− SEM are shown; each dot represents a sample. Mann–Whitney tests were used. * *p* < 0.05.

**Figure 4 cells-13-01458-f004:**
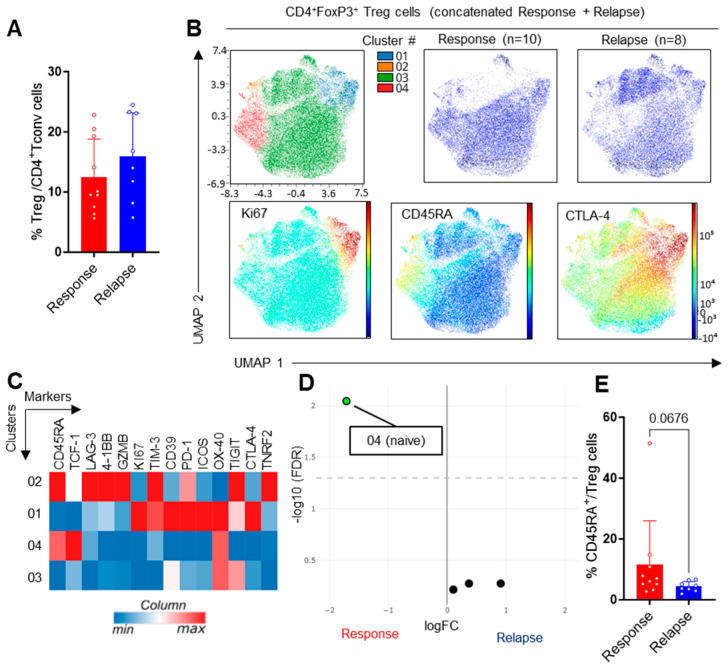
Analysis of Treg cells. (**A**) Proportion of Foxp3^+^ regulatory T cells (Treg cells) within CD4^+^ T cells following manual gating. (**B**) UMAP visualization, FlowSOM distribution of clusters and projection of selected markers in concatenated samples following manual gating on live Treg cells. (**C**) Heatmap showing hierarchical clustering and expression of indicated markers in FlowSOM clusters. (**D**) Volcano plot showing differential cluster enrichment. (**E**) Proportion of naive Treg cells upon manual gating. Means +/− SEM are shown; each dot represents a sample. Mann–Whitney tests were used.

**Figure 5 cells-13-01458-f005:**
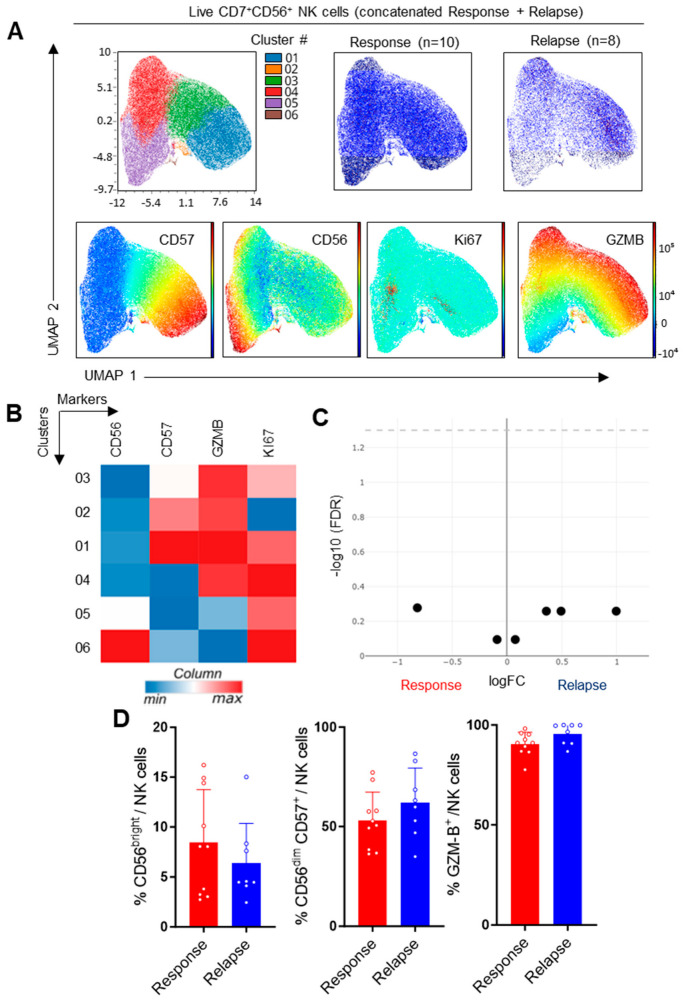
Analysis of NK cells. (**A**) UMAP visualization, FlowSOM distribution of clusters and projection of selected markers in concatenated samples following manual gating on live CD7^+^CD56^+^ NK cells. (**B**) Heatmap showing hierarchical clustering and expression of indicated markers in FlowSOM clusters. (**C**) Volcano plot showing differential cluster enrichment. (**D**) Expression of the indicated markers upon manual gating. Means +/− SEM are shown; each dot represents a sample. Mann–Whitney tests were used. No statistical difference was detected.

**Figure 6 cells-13-01458-f006:**
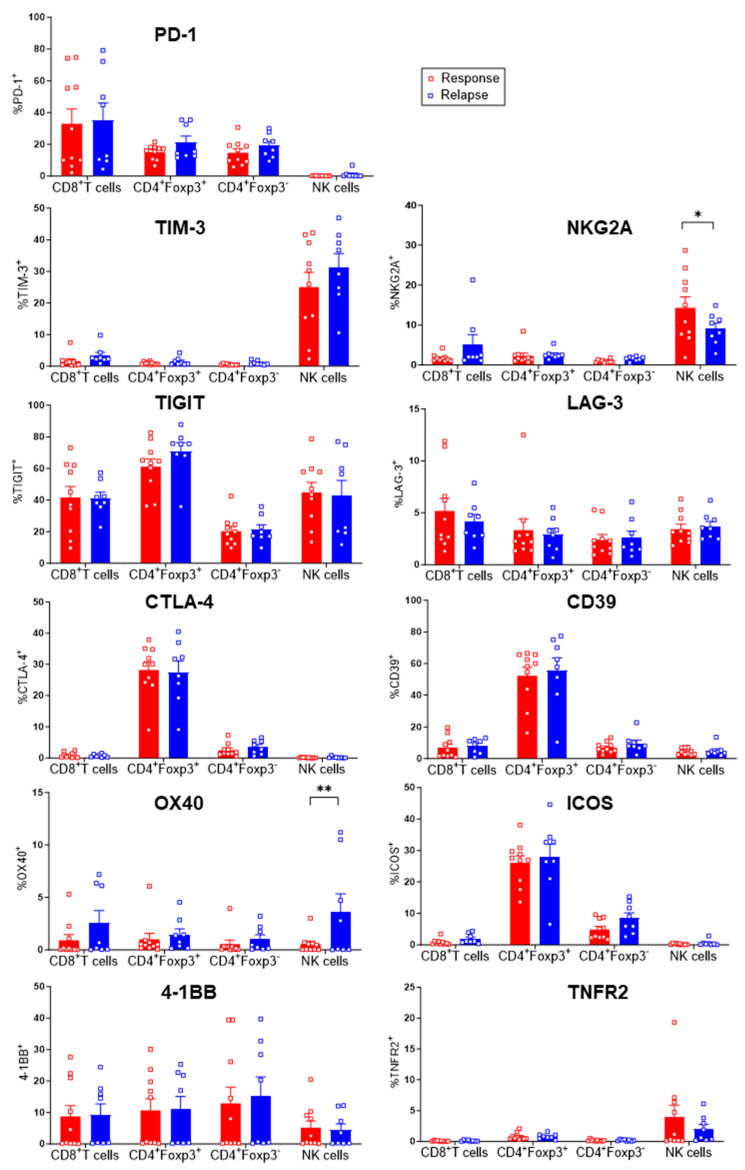
Expression patterns of checkpoint receptors in immune subsets. Proportion of checkpoint-expressing cells in different cell subsets following manual gating is shown as mean +/− SEM. Two-way ANOVA tests were used. * *p* < 0.05, ** *p* < 0.005.

**Table 1 cells-13-01458-t001:** Patient characteristics (*CR*: *complete response*).

Patient ID	Sex	Age (yrs)	TP53 Mutation (Yes/No)	Cytogenetic Status	IGHV Mutation Status	# of Previous Treatment Lines	Disease Status	Months under Ibrutinib Therapy
1	M	52	No	del11q	mutated	2	CR	84
2	M	73	Yes	del17p	unmutated	1	CR	83
3	M	68	No	None	unmutated	1	CR	36
4	F	77	No	None	unmutated	0	CR	36
5	F	68	No	None	unmutated	0	CR	60
6	M	72	No	None	unmutated	1	CR	48
7	F	80	No	None	unmutated	0	CR	24
8	F	67	No	None	unmutated	0	CR	24
9	M	79	No	None	unmutated	1	CR	24
10	F	62	No	None	unmutated	1	CR	48
12	F	84	Yes	del17p, complex karyotype	unmutated	1	Relapse	12
13	M	61	Yes	del17p, complex karyotype	unmutated	2	Relapse	36
14	F	45	No	del11q	unmutated	1	Relapse	48
16	M	76	No	None	unmutated	1	Relapse	24
17	F	89	No	None	unmutated	1	Relapse	36
18	M	51	No	None	unmutated	1	Relapse	72
19	M	67	Yes	del17p, complex karyotype	unmutated	0	Relapse	60
20	M	67	No	del11q	unmutated	1	Relapse	48

**Table 2 cells-13-01458-t002:** Patient characteristics at relapse (N/A: not assessed).

Patient ID	Disease Status	Months under Ibrutinib Therapy	BTK Mutation	Bulky Disease	Lymphocyte Count (G/L)	Months between Relapse Diagnosis and Sampling
12	Relapse	12	N/A	No	23.8	1
13	Relapse	36	C481S	No	13.3	2
14	Relapse	48	C481S	No	7	2
16	Relapse	24	N/A	No	20.39	5
17	Relapse	36	N/A	No	28.6	24
18	Relapse	72	C481S	No	54	1
19	Relapse	60	C481S	No	11.7	1
20	Relapse	48	None	No	8.22	1

## Data Availability

The original contributions presented in this study are included in the article/Appendix A; further inquiries can be directed to the corresponding author (Y.G.-B.).

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
