# Peer review of "Immunophenotyping of Peripheral Blood Cells in Patients with Chronic Lymphocytic Leukemia Treated with Ibrutinib"

_cells, 2024, doi:10.3390/cells13171458_

Round 1

Reviewer 1 Report

Comments and Suggestions for Authors

The Authors used high-dimensional spectral flow cytometry to analyze the phenotype of the peripheral blood immune cells of CLL patients receiving ibrutinib (10 responders, 8 progressed). Ibrutinib-resistant patients showed an increase in cycling CD8+ T cells and PD-1+ICOS+ TFH-like cells, whereas the expression of surface checkpoint receptors did not differ according to the ibrutinib response.

As the authors outlined, future research focused on the microenvironment of bone marrow and lymph nodes, as well as functional studies, is warranted to better define the role of the tumor microenvironment in CLL progression and its potential target of immunotherapy.

Major comments

1.    The small sample of patients analyzed does not make it possible to draw conclusive results regarding the immunophenotype profile associated with the response or resistance to ibrutinib of patients with CLL. The authors could expand the patient series to obtain more significant data than those presented in the current version of the paper.

2.    Baseline data on the phenotype of peripheral blood immune cells of CLL are lacking. The lack of this data does not allow to evaluate whether pre-existing characteristics of immune cells had an impact on the response to ibrutinib and whether treatment with ibrutinib had an effect in modifying the antigenic characteristics of T and NK cells.

Minor comments

1.    -Details about patients with CLL progression (BTK mutations, lymphocyte count, bulky disease could be of interest and should be reported.

2.    -Lines 239-241. The Authors stated:

“Despite this increased CD8+ T-cell activation upon relapse, we did not detect significant changes in the expression of inhibitory receptors, indicating that T-cell exhaustion, a feature of CLL rescued by Ibrutinib, was not (yet?) impacted by secondary progression.”

3.    The duration of treatment with on was highly variable, ranging between 12 and 84 months.

4.    Did the authors check whether the duration of ibrutinib therapy impacted the expression of inhibitory receptors?

5.    -Line 78: Please report the reference to the iwCLL guidelines.

-Line 23: Please change “complete pathological response” to “Complete response.”

Reviewer 2 Report

Comments and Suggestions for Authors

Pierre Stéphan and coauthors in this article analyzed by high-dimensional spectral flow cytometry the phenotype of non-B-cell immune populations in the circulating blood of CLL patients treated with Ibrutinib, displaying complete pathological response (CR) or secondary progression (Relapse). The aim of this study was to characterize the main immune circulating cell subpopulations to evaluate if it were possible to find any useful targets for immunotherapy in patients relapsing after treatment with ibrutinib. They included in their antibody panels 11 different surface checkpoints and cycling markers as well as various markers that identify functional subsets (e.g. regulatory and cytotoxic T cells).

They conclude that relapse upon ibrutinib does not lead to dramatic changes in the peripheral blood non-neoplastic immune cells even though they highlighted an increased proportion of cycling CD8+ T cell and of follicular T helper cells in the ibrutinib-resistant group at the expense of dendritic cells. Although these observations deserve further study for the characterization of the immunological profile of these patients, the experiment conducted produced data that deserve publication.

I just have a few observations:

1) did the authors characterize the presence of mutations in BTK, as this is known to be the major cause of resistance to Ibrutinb therapy?

2) what is the time point of the sampling of the CLL cases analyzed?

3) in the results paragraph from lanes 132 to 148 the reference to the figure does not correspond to the data described; for example, figure 1D may be figure 2D and references in the text to the panels 2C and 2E are missing

Reviewer 3 Report

Comments and Suggestions for Authors

In this work, Stéphan and colleagues used flow cytometry to compare the distribution and phenotype of non-B-cell immune populations in the circulating blood of CLL patients treated with Ibrutinib. Two groups were compared: patients with ongoing pathological response with those experiencing disease progression.

The Authors could not find any major changes in the composition of the immune cell subsets besides increased numbers of cycling of CD8+ T cells in progressive patients.  

When analyzing the expression of 11 different surface checkpoints on T-cell subsets, they found no difference in responding patients vs progressive ones.

As the Authors themselves mention in the Discussion, the major limitation of this study is the lack of analysis on longitudinal samples. Indeed, it is well known that the inter-patient variability is relevant, depending on different clinical and biological characteristics.

Moreover, the changes observed on the immune cell compartment during treatment response could simply be related to the tumor burden decrease. Accordingly, changes observed at disease progression might simply be secondary to tumor burden increase, rather than reflecting resistance to the specific drug ibrutinib. The increase in cycling CD8+ T cells and the reduction in naïve cell clusters in all T-cell subsets, as an example, is usually observed in active disease, irrespective of previous treatment. The Authors also comment on this issue in the Discussion, when writing “This suggests a rebound in the activation of tumor-specific immune cells due to the awaking of tumor burden, which was so far kept in check by the treatment”.

To rule out that the changes observed are merely secondary to disease progression, i.e. active disease, a direct comparison with the numbers and phenotype of the immune cells in the same patients before start of ibrutinib (i.e. when the disease was active but no treatment had yet been given) would be needed. I think that the analysis of baseline samples and therefore a longitudinal study would significantly add value to the paper.

In conclusion, I do not believe the paper much contributes to assess whether resistance to ibrutinib treatment is associated with alterations of the TME, which was the endpoint of the study.

Reviewer 4 Report

Comments and Suggestions for Authors

The research investigates how immune response is related to cancer outcome with a focus on immune therapies targeting T-cell inhibitory receptors in chronic lymphocytic leukemia (CLL). In CLL, there is altered immunophenotype characterized by increased CD 4+T cell activation and exhausted CD8+T cells. Ibrutinib shows in many pts initial success but often results in secondary resistance as well as affecting different subsets of immune cells. This study used spectral flow cytometry to phenotype immune cells in CLL patients who were either responding or relapsing to Ibrutinib. Its findings include the increase of CD8+ T cell frequencies at relapse and changes in CD4+ T cell subsets focusing on TFH- like cells. NK cells showed no significant differences between responders and relapsers. The study calls for further exploration into immune checkpoints and the necessity for functional assays so as to understand therapeutic consequences.

In this analysis, however, circulating blood has been considered alone leaving behind bone marrow and lymph node environments that are vital in case of CLL.

This study lacks functional assays which would have verified these phenotypic changes observed leading to no clear clinical implications.

The current study should have been longitudinal rather than cross-sectional so that more robust findings could be obtained about immune shifts over time.

While the research reveals variations in subsets of immune cells, no functional assays to assess T-cell cytotoxicity, cytokine production, and other immune functions to validate the phenotypic observations was carried out.

Sample size is small.

Round 2

Reviewer 3 Report

Comments and Suggestions for Authors

I suggest the Authors to exclude patient #17 from the analyses since the samples were taken 24 months after relapse and therefore are very unlikely to be representative of the disease status at relapse. By the way, didn´t the patient receive any other CLL treatment in between?

Author Response

Comment 1 : I suggest the Authors to exclude patient #17 from the analyses since the samples were taken 24 months after relapse and therefore are very unlikely to be representative of the disease status at relapse. By the way, didn´t the patient receive any other CLL treatment in between?

Response 1: We thank the reviewer for this suggestion. It is true that this patient was sampled long after ceasing the treatment. To address whether this sample may constitute an outlier in the relapsing cohort, we examined the distribution of immune cells and their phenotype. Reviewer Figure 1A shows that the global distribution of immune subset among live cells of patient 17, did not cluster separately from the other samples. Similar observations could be made for the general phenotype of CD8+ T cells, Tconv cells (Reviewer Figure 1B, C) and other subsets (not shown). We also assessed P17’s expression of immune checkpoints and found a pattern in line with other samples, although PD-1 expression was in the higher range, whereas CD39 expression in Treg cells was low. In light of these analyses, we decided to keep this sample in the study. This avoids re-running the whole unsupervised analysis pipeline, which would have changed all the Umap distributions, cluster identification and statistics across the paper. We hope the reviewer agrees with this decision.  

Of note, this patient received no additional therapy following Ibrutinib cessation, in particular because of her advanced age. She was kept under careful surveillance and is still alive to date.    

Reviewer 4 Report

Comments and Suggestions for Authors

Thank you for your replies,

Some of our comments and concerns were addressed, others could not be addressed because of the design of the trial and limitations of the scope of the paper/research project.

Author Response

Comment 1 : Thank you for your replies,

Some of our comments and concerns were addressed, others could not be addressed because of the design of the trial and limitations of the scope of the paper/research project.

Response 1: We thank again the reviewer for his/her thorough assessment of our paper, and hope the edited manuscript will bring new insights on immune responses in CLL, to the community.